# Investigation of NO Role in Neural Tissue in Brain and Spinal Cord Injury

**DOI:** 10.3390/molecules28217359

**Published:** 2023-10-31

**Authors:** Viacheslav V. Andrianov, Vladimir A. Kulchitsky, Guzel G. Yafarova, Leah V. Bazan, Tatiana K. Bogodvid, Irina B. Deryabina, Lyudmila N. Muranova, Dinara I. Silantyeva, Almaz I. Arslanov, Mikhail N. Paveliev, Ekaterina V. Fedorova, Tatiana A. Filipovich, Aleksei V. Nagibov, Khalil L. Gainutdinov

**Affiliations:** 1Zavoisky Physical-Technical Institute of the Russian Academy of Sciences, 420000 Kazan, Russia; slava_snail@yahoo.com (V.V.A.); gusadila@mail.ru (G.G.Y.); l_v_bazan@mail.ru (L.V.B.); 2Department of Human and Animals, Institute of Fundamental Medicine and Biology, Kazan Federal University, 420000 Kazan, Russia; tat-gain@mail.ru (T.K.B.); ira-kan@yandex.ru (I.B.D.); m.luda@mail.ru (L.N.M.); silantyevad@gmail.com (D.I.S.); arslanov-1999@mail.ru (A.I.A.); 3Brain Center, Institute of Physiology, National Academy of Sciences, 220012 Minsk, Belarus; vladi@fizio.bas-net.by (V.A.K.); katerina.minsk@mail.ru (E.V.F.); semionik88@mail.ru (T.A.F.); alexnag99@gmail.com (A.V.N.); 4Department of Biomedical Sciences, Volga Region State University of Physical Culture, Sport and Tourism, 420000 Kazan, Russia; 5Neuroscience Center, University of Helsinki, 00014 Helsinki, Finland; paveliev@outlook.com

**Keywords:** nitric oxide, cooper, brain stroke, spinal cord injury, electron paramagnetic resonance

## Abstract

Nitric oxide (NO) production in injured and intact brain regions was compared by EPR spectroscopy in a model of brain and spinal cord injury in Wistar rats. The precentral gyrus of the brain was injured, followed by the spinal cord at the level of the first lumbar vertebra. Seven days after brain injury, a reduction in NO content of 84% in injured brain regions and 66% in intact brain regions was found. The difference in NO production in injured and uninjured brain regions persisted 7 days after injury. The copper content in the brain remained unchanged one week after modeling of brain and spinal cord injury. The data obtained in the experiments help to explain the problems in the therapy of patients with combined brain injury.

## 1. Introduction

Nitric oxide (NO) serves as a vital signaling molecule that regulates cellular metabolism and various physiological functions in the human body [1,2,3,4,5]. The discovery of NO synthesis in mammalian cells has sparked extensive research efforts aimed at understanding its role in diverse fields of biology and medicine [2,6,7,8]. Endogenous NO production has been observed in a wide range of animal groups, plants, diatoms, slime molds, and bacteria [9]. NO exhibits ubiquitous presence within the nervous system [1,2], cardiovascular system [10,11], as well as other functional systems of the body, including immunity and metabolism [5,12,13]. NO also plays a significant role in many diseases, such as diabetes, cancer, ischemia, Alzheimer’s disease, and diseases of the cardiovascular system [14,15,16,17,18,19]. Among its crucial functions, NO plays a pivotal role in vasodilation [20,21]. Additionally, aside from its vasodilatory, neurotransmitter, and stress-limiting properties, NO has been implicated in oxidative stress reactions, the calcium glutamate cascade, and inflammatory processes [2,6,22,23,24].

There is ample evidence pointing to the fact that the disruption of nitric oxide biosynthesis is the primary factor of the brain pathophysiological response to hypoxia-ischemia [25,26,27]. The functional role of endogenous NO in the processes occurring during nervous system damage remains controversial and inadequately explored [4,5]. This is due to the fact that NO serves as a typical example of the classical two-faced Janus [28,29,30]. Thus, an increase in the activity of neuronal NO synthase (nNOS) was found at the beginning of ischemia with a maximum at the third hour [31], and the expression of inducible NOS (iNOS) was found a day after ischemia [32]. An overproduction of NO is also shown in acute hypoxia [33]. By using EPR spectroscopy, it was found that NO production increased after 5 min of ischemia and lasted for 60 min [34,35]. Fifteen minutes after ischemia, caused by occlusion of the middle cerebral artery, an increase in the relative concentration of free NO to 132% was found by using the method of injection of spin trap before ischemia [13,36]. On the other hand, a number of studies conducted on brain ischemia models have not confirmed the neurotoxic role of ischemically generated NO. For example, it has been found that NO-synthase inhibitors L-NNA and L-NAME do not reduce the size of lesions in focal cerebral ischemia in rats [37,38] but increase focal ischemic stroke [39]. There are numerous studies on the use of NO donors as neuroprotective agents after ischemic injury [40,41,42,43,44,45]. It has been found that short-term use of NO donors before the beginning of ischemia can protect the function of endothelium in ischemic reperfusion injury [46]. It was also shown that inhalation of NO or nitrite encourages brain function during perinatal hypoxic and chemical intervention [47].

Earlier, together with the team of Prof. V.B. Koshelev (Moscow State University), we conducted studies that aimed to investigate the content of NO in the focus of cerebral ischemia (left hemisphere) by using EPR spectroscopy. It showed that in the ischemic part of the left hemisphere cortex, the NO content in the spin trap decreased five times after 5 h of modeling an ischemic stroke, and this decrease persisted for a day after the stroke [48]. In another study, the authors of this article demonstrated that the formation of NO in the hippocampus decreased by 2–3 times 5 h after the symptoms of both ischemic and hemorrhagic stroke, and this decrease persists from 24 to 72 h [49]. It was found that NO content significantly reduced in the olfactory bulb of the rat brain after 1 and 2 days of modeling ischemia caused by carotid artery ligation [50]. The involvement of ATP-dependent K-channels in these processes has been shown [51,52].

Cerebral ischemia triggers the accumulation of excitatory amino acids in brain tissues and the activation of calcium-dependent nitric oxide synthase (NOS) isoforms, namely neuronal NOS and endothelial NOS. However, while selective inhibition of neuronal NOS demonstrates neuroprotective effects, selective inhibition of endothelial NOS produces neurotoxic outcomes [14,15,53,54,55]. Clinicians, who constantly take into account the individual reactions of each patient, are familiar with the paradoxical effects of NO. The reason for this fact is that the synthesis of NO by nNOS is accompanied by a calcium overload of the ischemic neuron caused by glutamate. It is assumed that the NO generated by eNOS expressed in endothelial cells, unlike nNOS, has a beneficial effect [27,45]. Here, eNOS not only promotes vasodilation but also increases the proliferation and migration of vascular smooth muscle cells, thereby enhancing arteriogenesis after a stroke. Inhibition of nNOS protected the brain from hypoxic-ischemic-induced injury in mice by the increased collateral flow. Therefore, the injection of a 7-nitroindazole blocker can be a potential utility for the treatment of ischemic reperfusion injury in human beings [56]. Such assumptions are supported by the results of a decrease in secondary brain injury after brain damage during inhalation of NO [57,58,59].

Disruption of cerebral oxygen supply also occurs in cases of thrombosis or an aneurysm of a blood vessel, and it often ends in the development of ischemic or hemorrhagic stroke [14,15]. The ambivalent role of NO is again manifested in the process of developing brain hypoxia-ischemia; according to modern concepts, NO can exert both neurotoxic and neuroprotective functions [14,28,43,51]. In response to the development of pathological processes of the brain hypoxia and ischemia, the regulatory systems of the brain, including the NO system, exhibit increased activity. This naturally leads to an elevation in oxygen consumption (which exacerbates hypoxia) and the accumulation of under-oxidized products in brain tissue, which, in aggregate, disrupts the integrative activity of the brain [60,61]. Despite the restoration of blood flow to the ischemic area, a large number of free radicals (both reactive nitrogen forms and reactive oxygen forms) are produced during reperfusion, which serves as important factors of ischemic reperfusion injury [55]. Thus, it is fundamentally important to take into account that the well-coordinated functioning of the NO system is disrupted with the development of hypoxia and cerebral ischemia. At the same time, cerebral ischemia is accompanied by multiple multidirectional changes in the NO content in the brain, which affects the functioning of the brain regulatory systems and the effectiveness of signal pathways to control the functions of all functional systems of the body [61,62,63]. Thus, there is increasing evidence that NO plays an important role in neuroprotection in stroke, even if NO is usually regarded as a toxic gas. Therefore, we need to be dialectical about NO, and further research, including animal studies and clinical studies, can give us a new understanding of the treatment of stroke and other diseases of the central nervous system [54].

Ischemia and trauma of the brain and spinal cord are characterized by primary injury and subsequent secondary phase of injury [58,64,65]. A critical component of secondary injury is oxidative stress and increased formation of reactive oxygen species [27,66]. Secondary injury occurs not only at the site of the initial primary injury but also leads to the spread of the lesion to neighboring, intact tissues. 

Activation of antioxidant enzymes represents a crucial defense mechanism against highly toxic oxygen radicals. The majority of these enzymes are associated with copper-containing proteins [67,68]. The one of them is Cu,Zn-superoxide dismutase (SOD) [68,69,70]. Various forms of SOD exist depending on the specific transition metal cofactor present in the active site of the enzyme. For instance, Cu,Zn-SOD features copper as the active site cofactor and zinc as a conformation-stabilizing cofactor, as well as a number of others [71]. The primary and essential defense mechanism against free radical oxidative processes involves the neutralization of superoxide radicals (O_2_^−^) by the cytosolic enzyme Cu,Zn-SOD [72]. This enzyme plays a significant role in the antioxidant protection of virtually all cells that come into contact with oxygen. Thus, maintaining a balanced copper level in the nervous system is crucial for its proper functioning. However, imbalances in copper have been implicated in the pathogenesis of numerous neurodegenerative disorders, including Parkinson’s, Alzheimer’s, and Huntington’s diseases, while disruptions in SOD metabolism can lead to various pathological conditions.

Brain injury following subarachnoid hemorrhage occurs in two phases: an acute ischemic stroke during the initial bleeding and subsequent secondary events, among which the most fatal is the cerebral vasospasm occurring 3–7 days later [73]. In order to correct and control the situation in clinical practice, clinicians need experimental information about the patterns of development of spasms of the blood vessels of the brain. In addition to spasms of blood vessels, intracranial injury can manifest as diffuse or focal damage of brain tissue and, to a certain extent, by a number of scientists and clinicians, refers to such volumetric pathologies, including but not limited to hematoma, cerebral aneurysm, tumor, and stroke [74,75]. Apart from the primary injury, secondary injuries are usually accompanied by a cascade over the following hours or days [73,75,76]. NO has been shown to play a multifaceted role in the regulation of cerebral blood flow both under normal physiological conditions and in pathology, for example, after a traumatic brain injury, with subarachnoid hemorrhage, severe traumatic brain injury, and ischemic stroke [15,77,78]. Several authors have highlighted the significance of NO as one of the triggers of the primary inflammatory pathway activated following hemorrhagic stroke [79].

Considering the aforementioned literature, we aimed to investigate not only the dynamics of nitric oxide production but also the copper content in both injured and non-injured regions of the frontal lobes of the brain, as well as the hippocampus, after modeling a combined brain and spinal cord injury. 

## 2. Results

EPR spectroscopy was employed to study the relative intensity of NO production and copper content (as an indicator of the first and third subunits of superoxide dismutase) in injured and non-injured (contralateral) areas of the brain (frontal lobe), as well as in the hippocampus after combined injury of the brain and spinal cord. The EPR spectrum of a sample from the frontal lobe of the rat brain is given in Figure 1. It showed the determination of the relative intensity of the signals from the Cu^2+^-(DETC)_2_ and (DETC)_2_-Fe^2+^-NO complexes with amplitudes equal to their contribution to the spectrum of the sample. The details of determining these parameters were provided in the experimental procedures section.

In Figure 2, the EPR spectrum of the intact-control (Control rat), injured (Injured rat, injured area), and non-injured (Injured rat, non-injured area) regions (frontal lobe) of the rat brain seven days after inducing a combined injury to the brain and spinal cord are shown. The EPR spectrum of NO appeared as a triplet between a magnetic field of 330–337 mT with a g-factor of 2.038 [80]. Additionally, a distinct EPR spectrum was observed from the Cu^2+^-(DETC)_2_ complex, representing the interaction of copper with diethyldithiocarbamate (DETC) [81,82,83]. The solid line represented the spectrum of the sample, while the dashed line corresponded to the signal of nitric oxide bound to the spin trap in the ((DETC)_2_-Fe^2+^-NO) complex spectrum (Figure 2). In the samples of biological tissues of rats that did not receive injections of spin trap components, lines characteristic of NO were not observed in the EPR spectrum of the complex ((DETC)_2_-Fe^2+^-NO). As an example, we have given the EPR spectrum of the liver, the signal from which is usually intense (Figure 3). The frames show the area of the magnetic field in which the EPR signal is observed ((DETC)_2_-Fe^2+^-NO) (Figure 2 and Figure 3). The relative changes in the amounts of the NO-containing complex and the Cu(DETC)_2_ complex were evaluated by the difference between the maximum and minimum of the first derivative of absorption signals from these complexes. The details of determining these parameters were given in the experimental procedures section.

Figure 4A displayed the statistical analysis of integral intensities for the (DETC)_2_-Fe^2+^-NO signal in the spectra of the investigated biological tissue samples obtained by EPR spectroscopy. This figure illustrated the spectral characteristics of both injured and non-injured (contralateral) brain areas after a combined injury of the brain and spinal cord, enabling the assessment of NO production in brain tissues. The results indicated a significant reliable decrease in NO production at 84% (*p* = 0.029, Mann–Whitney) in the injured brain area, as well as a significant but unreliable decrease in NO production at 66% (*p* = 0.38, Mann–Whitney) in the non-injured (contralateral) brain area, seven days after injury modeling. These findings demonstrated a distinct difference in NO production between the injured and contralateral brain areas (*p* = 0.05, Mann–Whitney). In the experimental data on the control samples, there were two data points close to 60; they were very different from the others. We carried out a processing option when excluding these two points from the statistical analysis (Figure 4B). There is a significant difference in a separate comparison between the group “Control rats” and group “Injured rats-injured area” by *t*-test (*p* = 0.021) and between the group “Injured rats-injured area” and group “Injured rats-non-injured area” (*p* = 0.05), and there is a significant difference when comparing all three groups by ANOVA test. In this case, a posteriori tests of different types showed a significant difference between these two groups (Figure 4B).

Additionally, Figure 4C presented statistical data on the integrated signal intensities of (DETC)_2_-Cu. The results revealed that the copper content remains unchanged in both the injured and non-injured brain areas seven days after injury modeling. In summary, the simulation of injury led to a significant decrease in NO production in both the injured and non-injured areas of the brain, while there were no observed changes in the copper content.

Figure 5 presented the EPR spectra of the hippocampal tissue of intact-control (Control rat) and injured (Injured rat) rats seven days after a combined injury to the brain and spinal cord. The spectroscopic characteristics of the complexes observed in our experiments closely resembled those obtained in previous studies [80,81,82]. The g-factor was determined to be g﬩= 2.025 for Cu^2+^-(DETC)_2_ and g = 2.035 for (DETC)_2_-Fe^2+^-NO. The EPR signal for this complex exhibits a triplet hyperfine structure. Similar data were previously obtained [84].

The results of spectrum analysis, as shown in Figure 6, indicated no significant (reliable) changes in NO production in the hippocampus following the modeling of a combined injury to the brain and spinal cord (an unreliable decrease of 34%). Furthermore, there was no observed alteration in the copper content, suggesting no changes in the activity of the antioxidant system. 

## 3. Discussion

The study of traumatic and ischemic brain injuries remains a prominent challenge in modern medicine [15,27,74,78,85]. The investigation of the reparative processes in nervous tissue and the development of innovative methods to restore neuronal structures are currently the most focal points in physiology and medicine. These research efforts hold substantial significance for advancing novel therapeutic and rehabilitation strategies [43,52,86].

Trauma and ischemia of the brain involve various pathological mechanisms that contribute to the disruption of nerve and glial cell integrity, as well as damage to blood vessels [15,87,88]. The shared similarities in certain stages of the pathogenesis between these cerebral lesions imply that therapeutic strategies aimed at protecting nervous tissue after ischemic events may also be applicable after brain injuries [54,60,89,90]. All these processes undergo significant changes when nervous tissue is damaged due to injury or stroke, whether of an ischemic or hemorrhagic nature.

Brain injuries disrupt the functioning of neural networks primarily due to mechanical damage to nervous tissue and blood vessels [54,91]. Apart from traumatic brain injury, dysfunctions of neurons and glia can develop due to compromised blood flow and the formation of hemorrhagic foci. In hemorrhagic strokes, the mechanical factor of nervous tissue compression in the cranial cavity is also present due to the formation of a hematoma. In ischemic strokes, brain tissue damage occurs due to hypoxia resulting from blood flow disturbances in the internal carotid arteries and/or the vertebrobasilar region [63]. Oxygen supply disruption to brain regions also occurs during blood vessel thrombosis or aneurysm rupture, which often leads to ischemic or hemorrhagic stroke [14,15]. During reperfusion, despite the restoration of blood flow to the ischemic area, a large number of free radicals (both reactive nitrogen forms and reactive oxygen forms) are produced, which serve as important factors of ischemic reperfusion injury [55].

In these processes of hypoxia-ischemia and mechanical brain damage, the role of NO appears to be contradictory, capable of both neurotoxic and neuroprotective functions [28,51]. Consequently, similarities are observed in the pathogenesis of brain injury and ischemic damage.

Using EPR spectroscopy, we investigated the relative intensity of NO production and the copper content in both injured and non-injured areas of the brain (frontal lobes), as well as in the hippocampus, during the simulation of a combined injury to the brain and spinal cord. These molecular components are of interest to researchers who are studying the mechanisms of brain function in normal and pathological conditions. Various methodological approaches are employed for experimental analysis, the EPR spectroscopy being one of the most sensitive available techniques [92,93]. Significantly, the electron spin trap method for detecting and quantifying NO in biological tissues, developed by Prof. A.F. Vanin and colleagues, has played a crucial role in advancing the EPR spectroscopy technique [94]. 

The experimental analysis of brain injuries revealed a notable decrease in NO production seven days after the injury simulation, both in the injured and non-injured (contralateral) areas of the brain. These results showed a distinct difference in NO production between the damaged and contralateral regions of the brain. In the experimental data on the control samples, there were two data points close to 60, which were different from the others. We carried out a processing option when excluding these two points from statistical analysis. There is a significant difference in a separate comparison between the group “Control rats” and group “Injured rats-injured area”) by *t*-test (*p* = 0.021) and between the group “Injured rats-injured area” and group “Injured rats-non-injured area” (*p* = 0.05) and when comparing all three groups by ANOVA test was shown. In this case, posteriori tests of different types showed a significant difference between these two groups.

In contrast, the copper content remained unchanged after seven days of injury modeling. Consequently, the trauma led to a significant decrease in NO production in both the injured and the contralateral intact brain area, while no alterations were observed in the activity of the antioxidant system. Notably, no changes in the NO production and copper content were observed in the hippocampus. This finding aligned with our previous measurements, where a brain injury was simulated using a different approach [78].

Considering the limited efficacy of current therapeutic approaches for brain injuries and strokes and ongoing debates regarding the reperfusion period, it is prudent to conduct more in-depth research on the mechanisms and therapeutic potential of existing approaches. Moreover, it is essential to take into account the experimental findings presented by the authors of the article. These results highlighted notable distinctions in the dynamics of nitrosyl stress and the status of antioxidant protection between brain trauma and hemorrhagic stroke. Such differences offered an experimental foundation for the development of novel comprehensive therapeutic tactics in this field of medicine.

## 4. Experimental Procedures

### 4.1. Animals

The study involved 20 male rats (*n* = 20) weighing between 200 and 300 g. The rats were housed in standard vivarium conditions with ad libitum access to food and water. The brain and spinal cord injury modeling was conducted at the Center of the Brain, Institute of Physiology, National Academy of Sciences of Belarus in Minsk. The experimental procedures followed the approved protocol of the Ethics Commission (Protocol No. 1, dated 31 January 2019; Ethic Committee Name: Ethics Commission of the Institute of Physiology, National Academy of Sciences of Belarus, Minsk; Approval Code: Approval code E7/04/2023; Approval Date: dated 31 January 2019) of the Institute of Physiology, National Academy of Sciences of Belarus, Minsk. Tissue samples from the injured and non-injured areas of the frontal lobes of the brain, as well as the hippocampus, were collected seven days after the surgery (*n* = 5), and the same number of animals (*n* = 5) were left to evaluate the effectiveness of restoration of central control of motor functions after surgery. Tissue samples taken from intact rats (*n* = 5) were also used as a control group, and 5 animals were left for assessment of motor functions. These dates were chosen for two reasons: on the one hand, it is the accounting of data in previously conducted experiments with immunohistochemical staining of damaged areas of the brain [95], and, on the other hand, it is behavioral experiments [96]. The biological samples were then transported from Minsk to Kazan in specialized containers. There were 20 animals left in Minsk after modeling brain and spinal cord injuries, which continued to be observed for a month after the start of the experiment to assess the effectiveness of restoring central control of motor functions.

### 4.2. Experiment Protocol: Modeling of Combined Trauma of the Brain and Spinal Cord in Rats

All surgical procedures were conducted on the anesthetized animals using a combination of ketamine (55.6 mg/kg), xylazine (5.5 mg/kg), and acepromazine (1.1 mg/kg), administered intraperitoneally [97].

Laboratory rats were anesthetized and secured in the prone position on the surgical table stretched by the limbs. The head of the animal was firmly secured, and the cranial vault area was prepared by removing hair and applying a 2% iodine solution to sterilize the skin. A midline incision of 10–12 mm was made along the cranial midline using a scalpel to access the underlying tissues. Locally, the periosteum was carefully dissected over the precentral gyrus, and a craniotomy was performed using a drill. The precentral region of the brain on the left hemisphere was then specifically targeted for localized brain tissue damage using a stylet. The procedure lasted approximately 3–4 min. Following the intervention, the incision was sutured, and the skin was treated again with a 2% iodine solution to minimize the risk of infection.

At the next stage, we continued to conduct a surgical procedure but were already at the lumbar spinal cord level. The hair in the lumbar region was removed, and the skin was sterilized using a 2% iodine solution. A longitudinal incision was made through the skin and soft tissues, aligning with the projection of the lumbar vertebrae. With utmost care, the stylet was inserted into the spinal cord at the level of the first lumbar vertebra. When the stylet was removed, the duration of wound bleeding was observed and noted in the diary. Once the bleeding ceased, the wound was closed with two sutures, and the skin was treated again with a 2% iodine solution.

Before the operation, a day after the operation, and a week (seven days) after the operation, pain thresholds (hot plate, tail-flick reflex) were determined in all animals.

After a period of one week (seven days) following the surgical procedure, tissue samples were extracted from both the injured area and the non-injured contralateral region of the brain (frontal lobe), specifically from the frontal lobe. Additionally, tissue samples were collected from the hippocampus. The control group of animals did not undergo surgical interventions and served as a baseline for comparison.

### 4.3. Formation of a (DETC)_2_-Fe^2+^-NO Complex with a Spin Trap in Rat Tissues 

The problem of quantitative determination of NO in living systems needs an unambiguous solution since binding all NO produced by living organisms with iron-dithiocarbamate traps is, in principle, impossible since part of the free NO molecules bypasses the traps. Nevertheless, their complete capture is theoretically possible and can be achieved through the use of ultra-high concentrations of NO traps in cells and tissues. However, this approach entails significant disturbances in cellular metabolism [98]. The major challenge in accurately determining the concentration of nitric oxide in its free form within tissues and body fluids is due to its highly reactive nature and short lifespan, resulting in low concentrations. Currently, the electron paramagnetic resonance (EPR) method is extensively employed for measuring NO production in biological systems [48,92,93]. EPR spectroscopy has emerged as one of the most reliable techniques for detecting and quantifying NO levels in biological tissues, especially when the technique of spin traps was proposed. In 1984, Vanin and colleagues proposed using a divalent iron complex with diethyldithiocarbamate (DETC) as a trap for NO in animal cells and tissues [94]. This method relies on the formation of a complex between Fe^2+^ and DETC to capture NO and generate a stable ternary complex, denoted as (DETC)_2_-Fe^2+^.

In this study, this spin trap technique was utilized for measurements. To form a spin trap, like previous experiments, DETC-Na was administered intraperitoneally at a dose of 500 mg/kg in 2.5 mL of water. Next, a solution mixture consisting of ferrous sulfate (FeSO_4_ × 7H_2_O, Sigma, St. Louis, MO, USA) at a dose of 37.5 mg/kg and sodium citrate at a dose of 187.5 mg/kg (in a volume of 1 mL of water per 300 g of animal weight) was prepared immediately before injection [49,84]. In a mixture of iron sulfate and sodium citrate, iron citrate was formed. DETC-Na and iron citrate were delivered to the cells with blood flow. A paramagnetic complex (DETC)_2_-Fe^2+^-NO, which was insoluble in water, was formed in the presence of NO (Figure 1). It was stable and could persist for a long time. The half-life of this molecule at room temperature is approximately 1.5 h [80]. DETC-Na and iron citrate were distributed throughout the body and formed a complex (DETC)_2_-Fe^2+^. Spin trap complex with NO (DETC)_2_-Fe^2+^-NO is characterized by an easily recognizable EPR spectrum with a g-factor value of g = 2.038 and three components of hyperfine structure [80,82,99,100].

In the experimental group, areas of brain damage and contralateral areas in the frontal lobes, as well as hippocampal tissue, were obtained after decapitation and trepanation by a few seconds. The tissue samples were then packed in an insulin syringe and placed in a container with liquid nitrogen. The inner diameter of the syringe matched the inner diameter of the finger Dewar, facilitating the measurement process of the tissue samples. Similar areas of the brain from rats weighing 100–200 mg in the control group were also collected and treated in the same manner for comparison.

### 4.4. Measurements of a (DETC)_2_-Fe^2+^-NO and Cu(DETC)_2_ Complexes in Rats Tissues 

The spectra of the biological sample (DETC)_2_-Fe^2+^-NO and Cu^2+^-(DETC)_2_ complexes were recorded using a Bruker X-band (9.5320 GHz) EMX/plus spectrometer. The sample in a Broker finger Dewar was placed in a 2-cavity double resonator (model ER 4105DR) at a magnetic field modulation frequency of 100 kHz, modulation amplitude of 2 G, microwave radiation power of 2 mW, time constant of 327 ms and temperature of 77 K. Throughout the experiments, the modulation amplitude, gain, and microwave power settings were carefully selected to ensure that overmodulation and saturation of the electron paramagnetic resonance signal were avoided. These parameters remained consistent across all measurements. The sample weight used for the experiments was approximately 100–200 mg. The amplitude of the EPR spectra was consistently normalized to the sample weight to ensure accurate comparison and analysis [48]. 

The finger Dewar sample was placed in the first cavity from a 2-cavity double resonator, and the reference sample was placed in another cavity of the same resonator. Since each test sample was under the same conditions as the reference sample, this made it possible to quantify the intensity of the test samples and compare them in intensity with each other. 

Figure 1 demonstrates the EPR spectrum of the rat frontal lobe sample. This spectrum was obtained by microwave radiation of constant frequency with a broach of the magnetic field. Signals from different paramagnetic particles (complexes) were present in this spectrum. In the magnetic field area from 330 to 337 mT, there was an overlap of signals of the spin trap complex with NO (DETC)_2_-Fe^2+^-NO, which was characterized by an easily recognizable EPR spectrum with a g-factor value of g = 2.038 and three components of hyperfine structure. The g-factor is determined by the well–known formula [93,100]:hν = gßH
where the measurement parameters are ν—frequency (in our case, it is equal to 9.5320 GHz) and H—the magnitude of the magnetic field. For NO, we determined the g-factor by the point where the first derivative of the central component of the hyperfine structure (HFS) intersects with the zero line. In the observed range of the magnetic field, there is an EPR spectrum from the Cu^2+^-(DETC)_2_ complex with a g-factor value of 2.04. As known, the spectra of this complex split into four components of the HFS [35,81,82,101]. 

The integral intensity of the EPR spectrum is measured by the area under the absorption line [93,100]. For evaluation measurements, in most experimental studies, the maximum value of the function (curve) along the ordinate axis is the amplitude intensity. We have applied such a modification of the method in our work.

The relative signal intensity from complexes (DETC)_2_-Fe^2+^-NO and Cu^2+^-(DETC)_2_ in the observed spectrum of the sample was determined sequentially. First, the relative intensity of the EPR spectrum of the complex Cu^2+^-(DETC)_2_ was calculated by the amplitude of its extreme high-field component of the hyperfine structure (HFS) (as the difference between the minimum and maximum of the first derivative of the absorption line) [35,99]. Then, the EPR spectrum of the Cu^2+^-(DETC)_2_ complex with a previously determined amplitude was subtracted from the EPR spectrum of the sample. Thus, we got rid of the contribution of the signal from the Cu^2+^-(DETC)_2_ complex in the area where the signal from the complex (DETC)_2_-Fe^2+^-NO is observed. Then, the relative intensity of the EPR spectrum of the complex (DETC)_2_-Fe^2+^-NO was determined using the measurements of its central component of the HFS (as the difference between the maximum and minimum of the first derivative of the absorption line). 

### 4.5. Statistical Processing of the Result

The data are presented as the mean ± standard error of the mean (M ± m). Statistical analysis was performed using the Student’s *t*-test and Mann–Whitney test. Differences were considered significant at *p* < 0.05.

## Figures and Tables

**Figure 1 molecules-28-07359-f001:**
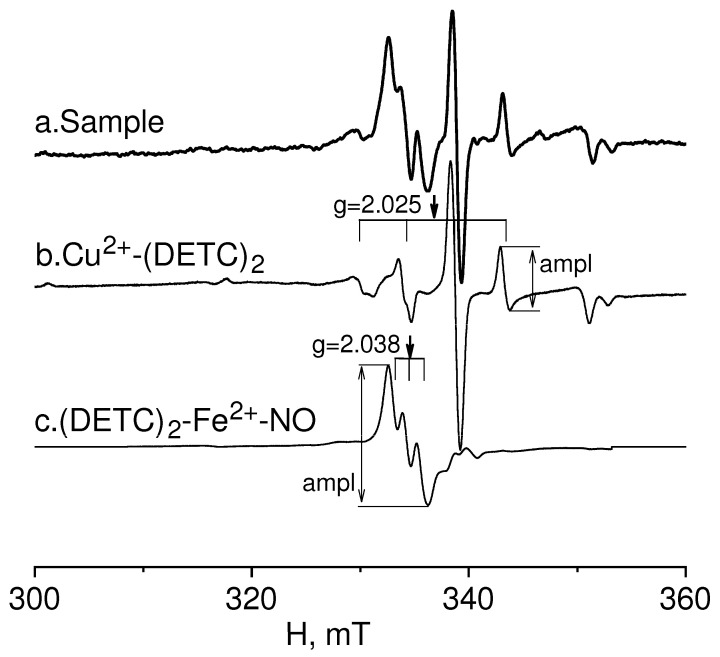
EPR spectrum of rat frontal lobe tissue (Sample-example) (**a**). Determination of the signal from the Cu(DETC)_2_ (**b**) complex and (DETC)_2_-Fe^2+^-NO (**c**) complex with amplitudes equal to their contribution to the spectrum of the sample.

**Figure 2 molecules-28-07359-f002:**
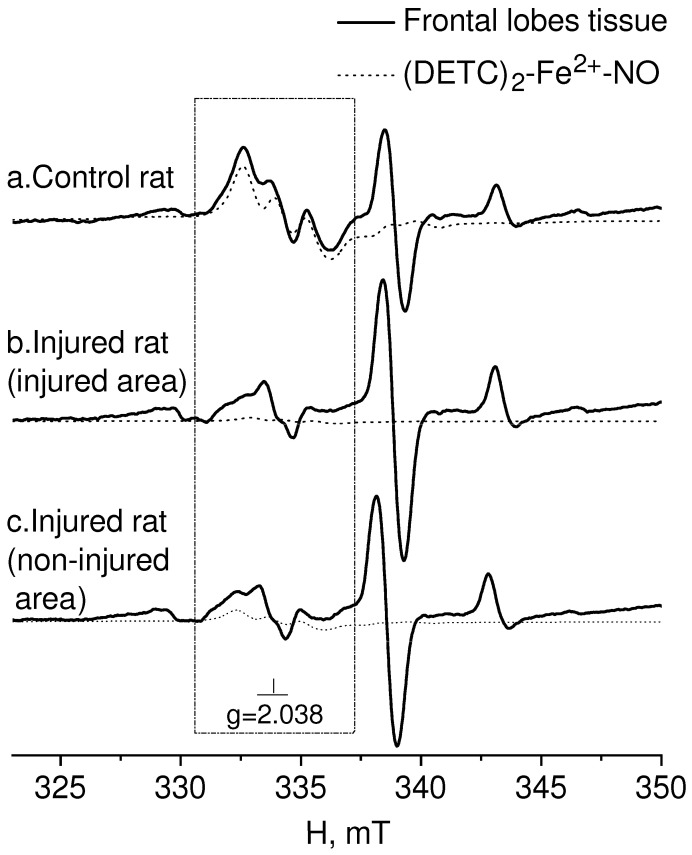
EPR spectrum of frontal lobe tissues from the control-intact (Control rat) and injured (Injured rat) rat: injured (injured area) and non-injured (non-injured area) regions of the rat brain 7 days after modeling a combined injury of the brain and spinal cord. The spectrum displays signals from (**a**) tissue sample, (**b**) (DETC)_2_-Fe^2+^-NO complex, and (**c**) Cu(DETC)_2_ complex. The calculated spectrum of the (DETC)_2_-Fe^2+^-NO complex is also depicted in the observed spectrum. The dotted line represents the (DETC)_2_-Fe^2+^-NO complex contribution to the observed signal. The frame shows the magnetic field area for the EPR spectrum of the (DETC)_2_-Fe^2+^-NO complex. The temperature is 77° K. The rats were injected with (DETC)_2_-Fe^2+^-citrate.

**Figure 3 molecules-28-07359-f003:**
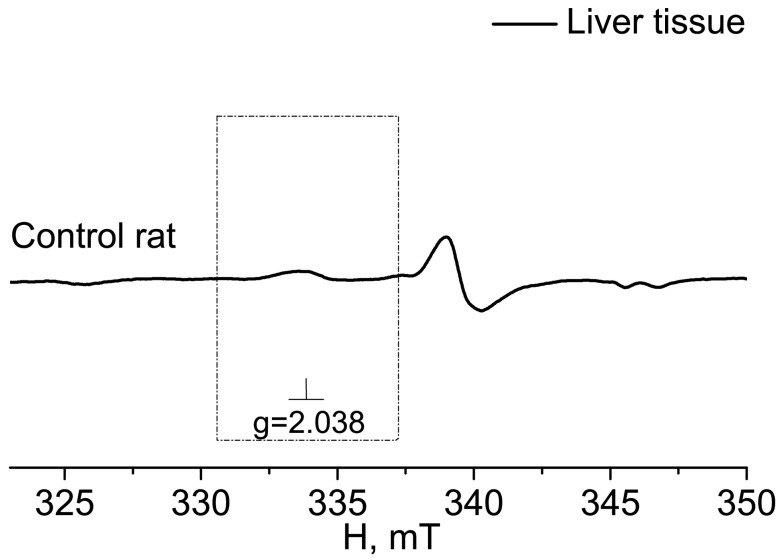
EPR spectrum of liver from the control (Control rat). The dotted rectangle shows the magnetic field area, which is characteristic of the spectrum EPR of (DETC)_2_-Fe^2+^NO. The frame shows the magnetic field area for the EPR spectrum of the (DETC)_2_-Fe^2+^-NO complex. The temperature is 77° K. The rats were injected with (DETC)_2_-Fe^2+^-citrate.

**Figure 4 molecules-28-07359-f004:**
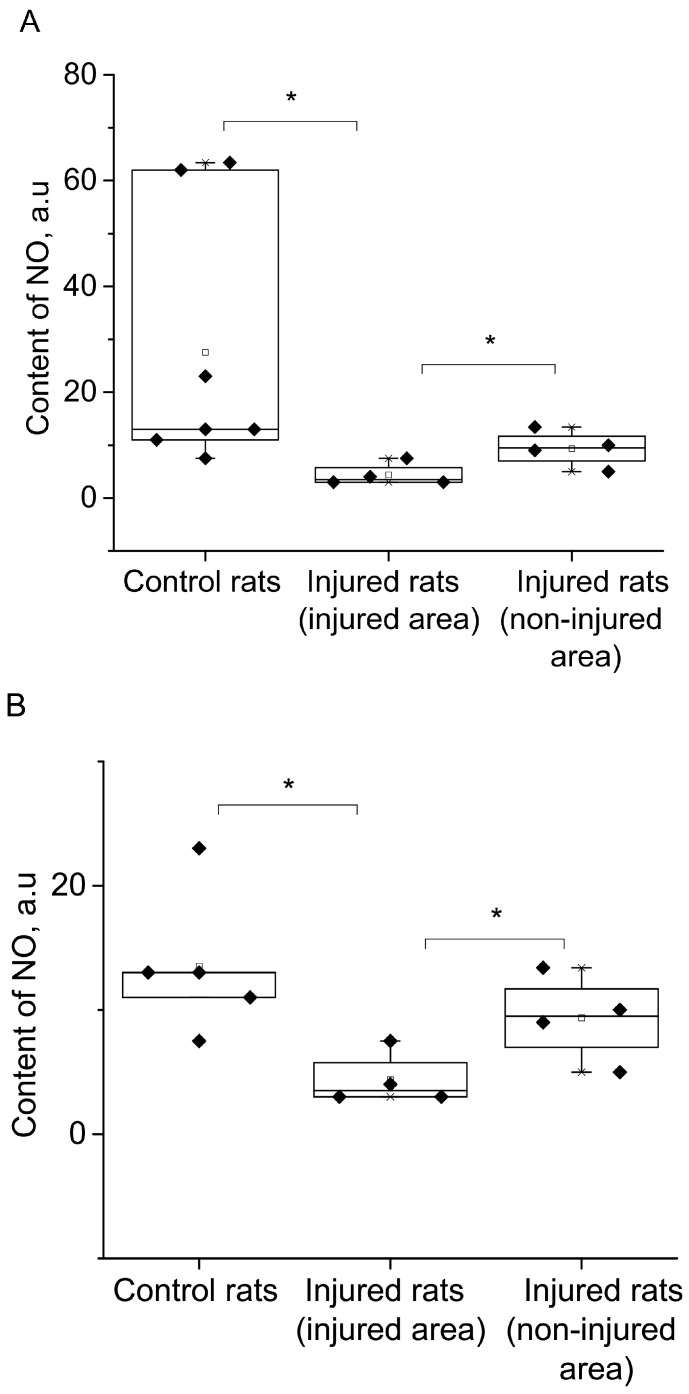
The content of NO (**A**,**B**) and Cu (**C**) in the tissues of the frontal lobes of the control (Control rats) and injured (Injured rat) rats: injured (injured area) and non-injured (non-injured area) regions of the rat brain 7 days after the combined injury of the brain and spinal cord. The average integral intensity of the (DETC)_2_-Fe^2+^-NO and Cu(DETC)_2_ signal is shown in %. (*) indicates a significant difference in NO content between the Control and Injured groups (*p* < 0.05).

**Figure 5 molecules-28-07359-f005:**
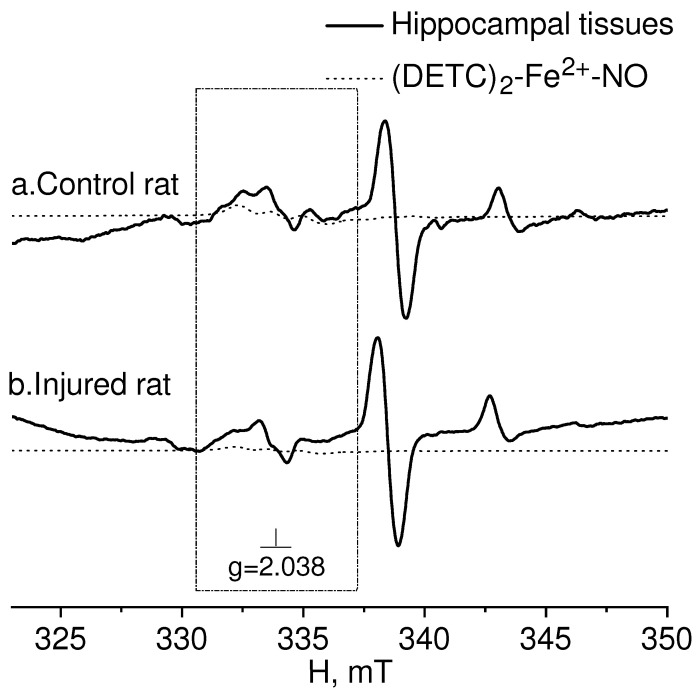
EPR spectrum of hippocampal tissues from the control (Control rat (**a**)) and injured (Injured rat (**b**)) rats 7 days after a combined injury to the brain and spinal cord. The signals of the tissue sample and the calculated spectrum of the (DETC)_2_-Fe^2+^-NO complex in the observed spectrum are shown. The dotted line represents the (DETC)_2_-Fe^2+^-NO complex contribution to the observed EPR spectrum. The frame shows the magnetic field area (DETC)_2_-Fe^2+^-NO complex. The temperature is 77° K. The rats were injected with (DETC)_2_-Fe^2+^-citrate.

**Figure 6 molecules-28-07359-f006:**
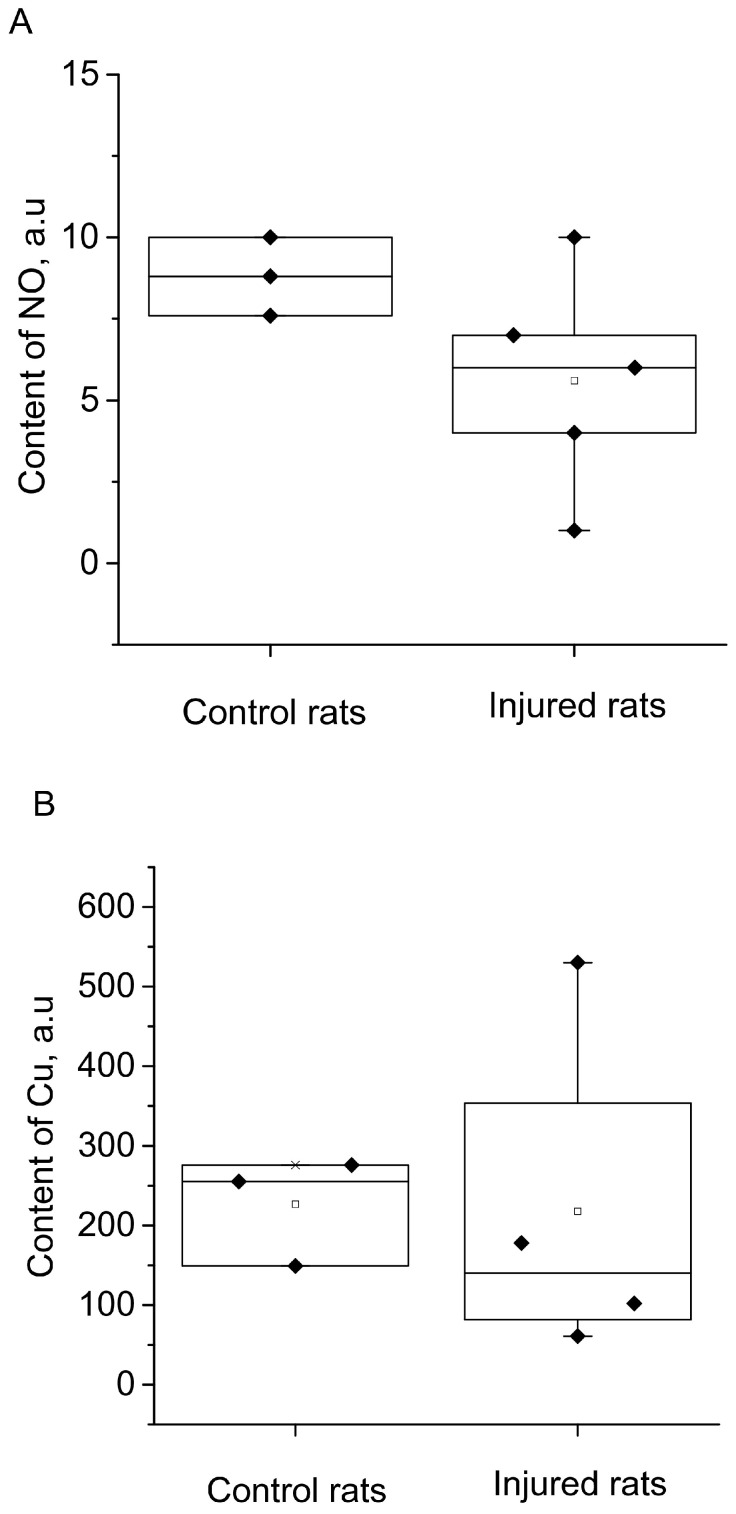
The content of NO (**A**) and Cu (**B**) in the hippocampus of the control (Control rats) and injured (Injured rats) rats 7 days after the combined injury of the brain and spinal cord. The average integral intensity of the (DETC)_2_-Fe^2+^-NO and Cu(DETC)_2_ signals is also shown in the figure (R.u.).

## Data Availability

The data confirming the stated results can be found in the Brain Center of the Institute of Physiology of the National Academy of Sciences of Belarus and the Laboratory of Spin Physics and Spin Chemistry of the Kazan Institute of Physics and Technology of the Russian Academy of Sciences.

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
