# Peer review of "Investigation of NO Role in Neural Tissue in Brain and Spinal Cord Injury"

_molecules, 2023, doi:10.3390/molecules28217359_

Round 1

Reviewer 1 Report

Comments and Suggestions for Authors

This paper is in a continuing series of related studies by the authors. It is interesting to get a report from this group on a slightly different  Possibly they have forgotten which of their prior papers they should cite for crucial information.  The papers they cited (refs. 53, 54) do not provide the needed information about how they assessed the EPR spectra.  Nothing is said in any of the prior papers that I scanned about how they used EPR to measure copper.  None of the papers identify any of the EPR signals other than the 3-line pattern due to the Fe complex with NO.

 What is the explanation for the very large gap in the NO levels in the control group?  It seems that all of the interpretation of the paper depends on the 2 data points near 60.

 The paper needs to be revised to explain the EPR spectra, explain how they measured copper, and explain the distribution in NO values in the control group.

Comments on the Quality of English Language

I did not find problem reading the text

Author Response

An EPR Spectroscopy Study of NO and Copper Content in Injured and Uninjured Areas of the Brain After a Combined Injury of the Brain and Spinal Cord in Rats

Authors

Viacheslav V.Andrianov, Vladimir A.Kulchitsky, Guzel G.Yafarova, Leah V. Bazan, Tatiana K. Bogodvid, Irina B. Deryabina, Lyudmila N. Muranova, Dinara I. Silantyeva, Almaz I. Arslanov, Mikhail N. Paveliev, Ekaterina V. Fedorova, Tatiana A. Filipovich, Aleksei V. Nagibov, Khalil L.Gainutdinov*

Dear Reviewer,

Thank You for the review. We consider all Your comments.

We are grateful for the careful reading of our manuscript and its discussion. You also highlighted the main results obtained by us, which seem significant in this scientific direction.

  1. This paper is in a continuing series of related studies by the authors. It is interesting to get a report from this group on a slightly different Possibly they have forgotten which of their prior papers they should cite for crucial information. The papers they cited (refs. 53, 54) do not provide the needed information about how they assessed the EPR spectra.  Nothing is said in any of the prior papers that I scanned about how they used EPR to measure copper.  None of the papers identify any of the EPR signals other than the 3-line pattern due to the Fe complex with NO.

Response: Thank you for statement of this question. Indeed, we understood that in the cited articles we described only the general course of evaluation of EPR spectra. In response to this remark, we have added a detailed description of the process of estimating the parameters of the EPR spectrum, which related to both NO and copper. I wanted to cite excerpts from the section "Experimental procedures" devoted to the issue of estimating EPR spectra:

«EPR spectroscopy has emerged as one of the most reliable techniques for detecting and quantifying NO levels in biological tissues, especially when the technique of spin traps was proposed. In 1984, Vanin and colleagues proposed using a divalent iron complex with diethyldithiocarbamate (DETC) as a trap for NO in animal cells and tissues.»

«The spectrum of the biological samples complexes were recorded using a Bruker X-band (9.5320 GHz) EMX/plus spectrometer. The sample in a Broker finger Dewar was placed in a 2-cavity of double resonator (model ER 4105DR) at a magnetic field modulation frequency of 100 kHz, modulation amplitude of 2 G, microwave radiation power of 2 mW, time constant of 327 ms and temperature of 77° K.»

«Figure 1 demonstrated the EPR spectrum of the rat frontal lobe sample. Signals from different paramagnetic particles (complexes) were present in this spectrum. In the magnetic field area from 330 to 337 mT, there was an overlap of signals of the spin trap complex with NO (DETC)2-Fe2+-NO, which was characterized by an easily recognizable EPR spectra with a g-factor value of g=2.038 and a triplet hyperfine structure, as well as the second line of the Cu2+-(DETC)2 complex with a g-factor value of 2.04. As known, the spectra of this complex was represented by four lines [81,82,102, 103]. This signal was present in our measurements in the field range from 335 to 343 mT.»

«Signal intensity from complexes (DETC)2-Fe2+-NO and Cu2+-(DETC)2 was determined sequentially. First, the intensity of the Cu2+-(DETC)2 signal was calculated along its right most line by the difference between the minimum and maximum of the line signal [71, 73]. Then, the signal from the Cu2+-(DETC)2 complex was subtracted from the observed spectra with the intensity found in this way. Thus, we got rid of the contribution of the signal from the Cu2+-(DETC)2 complex in the area where the signal from the complex (DETC)2-Fe2+-NO is observed. Then the signal intensity (DETC)2-Fe2+-NO was determined by the difference between the maximum of the first line and the minimum of the third line of the triplet in the remaining spectrum. By this way the signal intensity from the Cu2+-(DETC)2 complex was calculated first, and then adjusted for this signal, the signal intensity from the complex (DETC)2-Fe2+-NO was calculated.»

2.In addition, in response to reviewer 2, we gave an example of the EPR spectrum obtained without the use of spin traps (Fig. 3).

  1. What is the explanation for the very large gap in the NO levels in the control group?  It seems that all of the interpretation of the paper depends on the 2 data points near 60.

Response: We could not explain the large gap in NO levels in the control group. But we wanted to say that usually the point gap in the control groups was larger than in the experimental groups. In future, we are going to carry out additional measurements of samples from the brain of rats, as part of experiments in Minsk. Unfortunately, the characteristics of rats in Kazan differ from those in Minsk. Otherwise, we would be able to answer this question experimentally at once.

We have thought about this question ourselves and compared different groups according to two criteria:

«The results indicated a significant reliable decrease in NO production at 84% (P = 0.029, Mann-Whitney) in the injured brain area, as well as a significant but unreliable decrease in NO production at 66% (P = 0.38, Mann-Whitney) in the non-injured (contralateral) brain area, seven days after injury modeling. These findings demonstrated a distinct difference in NO production between the injured and contralateral brain areas (P = 0,05, Mann-Whitney). In the experimental data on the control samples, there were two data points close to 60; they were very different from the others. We carried out a processing option when excluding these two points from the statistical analysis (Fig. 4B). In both variants, there is a significant difference, as in a separate comparison of the two groups ("Control rats", "Injured rats-injured area") by t-test (P = 0.021), and when comparing all three groups by ANOVA test. In this case, a posteriori tests of different types showed a significant difference between these two groups (Fig. 4B).»

We thank you for a thorough reading of the article and suggestions for improving the quality of work. All comments are taken into account in the new text of the article.

Sincerely,

Prof. Kh.L. Gainutdinov, Lieder scientist of Laboratory of Spin Physics and Spin Chemistry of Zavoisky Physical-Technical Institute of the Russian Academy of Sciences, professor of Kazan Federal University.

Reviewer 2 Report

Comments and Suggestions for Authors

1. Title 

The end of the title needs to be changed ("after modeling" is not correct, did the authors mean that this is a model system). Also, hippocampus is a part of the brain, so it cannot be "brain and hippocampus".

2. Abstract

Same comment as for the Title about the phrase "after modeling".

Clarify "brain regions".

3. Line 62, Cyt c-oxidase is mentioned here but never discussed later in the text, only SOD. 

4. Line 65, What does it mean less common form, where? Also, what about Mn?

5. Introduction - The authors need to clarify better how a physical injury contributes to ROS production (and use appropriate references) and then go on explaining the activity of SOD. Also, is 7 days enough to see all changes. What about secondary injury (please see Jia, Z. et al, Spinal Cord 50, 264–274 (2012). https://doi.org/10.1038/sc.2011.111)

6. Line 91, text in the parenthesis is not clear

7. Line 92, same comment as before about "modeling"

8. Line 94, correct show to shows

9. The authors need to show the recorded EPR spectra of the controls: the Fe-(DETC)2-NO complex and the Cu-(DETC)2-NO complex. Furthermore, it would be nice to see room temperature spectra, the contribution from NO would be more obvious. 

10. Line 136, How are these statements corroborated, what are the values of g, A, and line widths of the signals shown in Figure 1.

11. The dotted line in Figure 1 is marked as the DETC-Fe-NO complex, but here it says that it is the contribution of NO, please be more specific.

12. Lines 141 and 159: In the Materials and methods section it says that the rats were injected separately with DETC and iron citrate, and here it says otherwise. Also, how are you sure that DETC reacts with the injected iron salt and not with other divalent metals in the body, Cu, Mn, Zn, and Fe.

13. Line 290, the ER 4112HV is a Helium cryostat, why was the finger dewar used. How was it even inserted in the cryostat.

14. The power of 30 mW seems strange for two reasons, 1-it is too high even for a room temperature experiment, and should be at least 10 times smaller for low T measurements; 2- the finger dewar produces very bad S/N, at such high power, the noise would be extreme and the spectra which are shown in the manuscript have very low noise.

15. The authors need to explain better how they measured the Cu content, and which results lead to their conclusion how there is no change in the Cu content.

Comments on the Quality of English Language

Minor editing of English language required

Author Response

An EPR Spectroscopy Study of NO and Copper Content in Injured and Uninjured Areas of the Brain After a Combined Injury of the Brain and Spinal Cord in Rats

Authors

Viacheslav V.Andrianov, Vladimir A.Kulchitsky, Guzel G.Yafarova, Leah V. Bazan, Tatiana K. Bogodvid, Irina B. Deryabina, Lyudmila N. Muranova, Dinara I. Silantyeva, Almaz I. Arslanov, Mikhail N. Paveliev, Ekaterina V. Fedorova, Tatiana A. Filipovich, Aleksei V. Nagibov, Khalil L.Gainutdinov*

Dear Reviewer,

Thank You for the review. We consider all Your comments.

We are grateful for the careful reading of our manuscript and its discussion. You also highlighted the main results obtained by us, which seem significant in this scientific direction.

  1. Title. The end of the title needs to be changed ("after modeling" is not correct, did the authors mean that this is a model system). Also, hippocampus is a part of the brain, so it cannot be "brain and hippocampus".
  2. Abstract. Same comment as for the Title about the phrase "after modeling". Clarify "brain regions".

Response: We agree with your remark that the term "modeling" suitable for general discussion, and in the text and title of article it is necessary to designate a specific experiment. We changed the name as well as the phrase in the abstract: «An EPR Spectroscopy Study of NO and Copper Content in Injured and Uninjured Areas of the Brain After a Combined Injury of the Brain and Spinal Cord in Rats».

We investigated NO changes in the injured and non-injured areas of the frontal lobes of the brain of male Wistar rats.

3-4.

We removed other copper-containing enzymes from the description, leaving only SOD.

  1. Introduction - The authors need to clarify better how a physical injury contributes to ROS production (and use appropriate references) and then go on explaining the activity of SOD. Also, is 7 days enough to see all changes. What about secondary injury (please see Jia, Z. et al, Spinal Cord 50, 264–274 (2012). https://doi.org/10.1038/sc.2011.111)

Response: We have significantly expanded the introduction with a description of the role of NO and ROS. These dates were chosen for two reasons: on the one hand, it is the accounting of data in previously conducted experiments with immunohistochemical staining of damaged areas of the brain [Shanko et al, Mechanisms of Neural Network Structures Recovery in Brain Trauma Biomed. J. Sci. Tech. Res. 7 (5) MS.ID.001567 (2018)]. And, on the other hand, the behavioral experiments showed basic effects at this time [Bogodvid et al, Effect of intranasal administration of mesenchymal stem cells on the approximate motor activity of rats after simulation of ischemic stroke. Eur. J. Clin. Investig. 49 (Suppl 1, P146-T), 161 (2019)].

6,7,8. We have corrected these comments.

  1. The authors need to show the recorded EPR spectra of the controls: the Fe-(DETC)2-NO complex and the Cu-(DETC)2-NO complex. Furthermore, it would be nice to see room temperature spectra, the contribution from NO would be more obvious. 

Response: We added a recording of the liver EPR spectrum (there is always more signal there) obtained without a spin trap in the article.

  1. Line 136, How are these statements corroborated, what are the values of g, A, and line widths of the signals shown in Figure 1.
  2. The dotted line in Figure 1 is marked as the DETC-Fe-NO complex, but here it says that it is the contribution of NO, please be more specific.

Response: We have added Figure 1, which showed the measurement of the intensities of the studied complexes. There you can also see how the signal from NO is allocated.

  1. Lines 141 and 159: In the Materials and methods section it says that the rats were injected separately with DETC and iron citrate, and here it says otherwise. Also, how are you sure that DETC reacts with the injected iron salt and not with other divalent metals in the body, Cu, Mn, Zn, and Fe.

Response: These methods have been studied in detail and elaborated by the authors of this method (Mikoyan, V.D.; Kubrina, L.N.; Serezhenkov, V.A.; Stukan, R.A.; Vanin, A.F. Complexes of Fe2+ with diethyldithiocarbamate or N-methyl-D-glucamine dithiocarbamate as traps of nitric oxide in animal tissues Biochim. Biophys. Acta 1997, 1336, 225-234).

  1. Line 290, the ER 4112HV is a Helium cryostat, why was the finger dewar used. How was it even inserted in the cryostat.

Response: This is our mistake. Previously, we carried out measurements on another ER 200 SRC spectrometer. These parameters were entered into the article from there. (The spectrum of the biological samples complexes were recorded using a Bruker X-band (9.5320 GHz) EMX/plus spectrometer. The sample in a Broker finger Dewar was placed in a 2-cavity of double resonator (model ER 4105DR) at a magnetic field modulation frequency of 100 kHz, modulation amplitude of 2 G, microwave radiation power of 2 mW, time constant of 327 ms and temperature of 77° K.)

  1. The power of 30 mW seems strange for two reasons, 1-it is too high even for a room temperature experiment, and should be at least 10 times smaller for low T measurements; 2- the finger dewar produces very bad S/N, at such high power, the noise would be extreme and the spectra which are shown in the manuscript have very low noise.

Response: We have corrected this error (microwave radiation power of 2 mW).

  1. The authors need to explain better how they measured the Cu content, and which results lead to their conclusion how there is no change in the Cu content.

Response: We have given a detailed description of the content measurement in the methodological part of the article.

All these comments are taken into account in the new text of the article.

Sincerely,

Prof. Kh.L. Gainutdinov, Lieder scientist of Laboratory of Spin Physics and Spin Chemistry of Zavoisky Physical-Technical Institute of the Russian Academy of Sciences, professor of Kazan Federal University.

Round 2

Reviewer 2 Report

Comments and Suggestions for Authors

Major: 

English language/grammar must be corrected prior to publication. I suggest to have the manuscript proofread by a native speaker. In this form, the manuscript is unaccaptable for publication, and also for peer review. There are phrases which are completely incorrect, most probably due to translation issues. 

For example:

Line 16, frontal lobes brain

Line 17 "using a stylet and hemorrhage injury" - the authors probably meant to say that the injury was made using a stylet, and that this caused a hemorrhage.

Line 65, "...we conducted studies aimed to study the content of NO in the focus of cerebral ischemia"

Line 123, Cu,ZnSOD is NOT famous, perhaps it is most studied, although I doubt it.

Additonally:

Figure 2. In the Figure legend the Cu complex is mentioned but the spectrum is not shown in the Figure.

My comments # 9, 10 and 11 were not amended. I don't understand why the authors added an EPR spectrum of the liver. It is not discussed in the manuscript, and I certainly did not ask for this.

My comment #15 was not amended, the manuscript is missing the part in which the authors show a clear connection between the results and the conclusions.

With regard to the authors answer to my comment #15, in Part 4.4 - what was the reference sample? Line 431 - "right most line" is not EPR terminology. Overall this explanation for signal intensity determination was not asked for in my comment #15,

In my opinion new Figure 7 is only suitable as a graphical abstract.

Title - Spectroscopic instead of Spectroscopy

Comments on the Quality of English Language

Extensive editing of English language required. English must be improved prior to further peer review.

Author Response

An EPR Spectroscopy Study of NO and Copper Content in Injured and Uninjured Areas of the Brain After a Combined Injury of the Brain and Spinal Cord in Rats

Authors

Viacheslav V.Andrianov, Vladimir A.Kulchitsky, Guzel G.Yafarova, Leah V. Bazan, Tatiana K. Bogodvid, Irina B. Deryabina, Lyudmila N. Muranova, Dinara I. Silantyeva, Almaz I. Arslanov, Mikhail N. Paveliev, Ekaterina V. Fedorova, Tatiana A. Filipovich, Aleksei V. Nagibov, Khalil L.Gainutdinov*

Dear Reviewer,

Thank You for the review. We consider all Your comments.

We are grateful to you for carefully reading our manuscript and discussing it. After the 1st review, you and the second reviewer asked a sufficient number of questions and made comments on various sections of our research. Therefore, several people of our teams answered them at once. When we combined all the corrections in the article, in several cases annoying errors occurred when we saved another version of the text. As a result, parts of the article appeared that did not quite fit together. We sincerely apologize for these errors.

Your comments:

Line 16, frontal lobes brain

We have replaced with: frontal lobes of the brain

Line 17 "using a stylet and hemorrhage injury" - the authors probably meant to say that the injury was made using a stylet, and that this caused a hemorrhage.

We have corrected and replaced with: Brain injury was performed by local destruction of the left precentral region of the brain using a stylet and local destruction of spinal cord using a stylet caused a hemorrhage injury was performed at the level of the first lumbar vertebra of the spinal cord.

Line 65, "...we conducted studies aimed to study the content of NO in the focus of cerebral ischemia"

We have replaced with: we conducted studies that aimed to investigate the content of NO in the focus of cerebral ischemia (left hemisphere) by using EPR spectroscopy.

Line 123, Cu,ZnSOD is NOT famous, perhaps it is most studied, although I doubt it.

We have replaced with: The one of them is Cu,Zn-superoxide dismutase

We fully agree with these comments, the answers to them lead to a clearer understanding of the text of the article.

Additonally:

Figure 2. In the Figure legend the Cu complex is mentioned but the spectrum is not shown in the Figure.

We agree with you that the mention of Cu complex in the legend is superfluous, it appeared there from another legend. We have removed this part of the legend.

My comments # 9, 10 and 11 were not amended. I don't understand why the authors added an EPR spectrum of the liver. It is not discussed in the manuscript, and I certainly did not ask for this.

  1. The authors need to show the recorded EPR spectra of the controls: the Fe-(DETC)2-NO complex and the Cu-(DETC)2-NO complex. Furthermore, it would be nice to see room temperature spectra, the contribution from NO would be more obvious. 

We probably misunderstood the first part of this question. Figures 1 and 2 show the spectra of the frontal lobes of control animals. Both complexes are clearly visible in these figures. Since we always have more signals from nitric oxide in the liver than in the nervous tissue, therefore, we proposed for consideration by the editors and reviewers the spectrum of the liver obtained in the absence of a spin trap. Since measurements with microwave radiation are carried out at the temperature of liquid nitrogen, we have not carried out such measurements at room temperature. In any case, there is no signal from nitric oxide without a spin trap.

  1. Line 136, How are these statements corroborated, what are the values of g, A, and line widths of the signals shown in Figure 1.

The g-factor is determined by the well–known formula [93,101] (Hogg, 2010; Kleschyov, 2007) hν = gßH, where the measurement parameters are v – frequency (in our case it is equal to 9.5320 GHz) and H - the magnitude of the magnetic field. For NO we determined the g-factor by the point where the first derivative of the central component of the hyperfine structure (HFS) intersects with the zero line.

For a complete answer to this question, I want to give an excerpt from the new edition of this part of the methods.

The integral intensity of the EPR spectrum is measured by the area under the absorption line [93,101]. For evaluation measurements, in most experimental studies, the maximum value of the function (curve) along the ordinate axis is the amplitude intensity. We have applied such a modification of the method in our work.

The relative signal intensity from complexes (DETC)2-Fe2+-NO and Cu2+-(DETC)2 in the observed spectrum of the sample was determined sequentially. First, the relative intensity of the EPR spectrum of the complex Cu2+-(DETC)2 was calculated by the amplitude of its extreme high-field component of the hyperfine structure (HFS) (as the difference between the minimum and maximum of the first derivative of the absorption line) [35,100]. Then, the EPR spectrum of the Cu2+-(DETC)2 complex with a previously determined amplitude was subtracted from the EPR spectrum of the sample. Thus, we got rid of the contribution of the signal from the Cu2+-(DETC)2 complex in the area where the signal from the complex (DETC)2-Fe2+-NO is observed. Then the relative intensity of the EPR spectrum of the complex (DETC)2-Fe2+-NO was determined using the mesurments of its central component of the HFS (as the difference between the maximum and minimum of the first derivative of the absorption line).

We have made these corrections in the "Experimental procedures" section

  1. The dotted line in Figure 1 is marked as the DETC-Fe-NO complex, but here it says that it is the contribution of NO, please be more specific.

We have corrected this inaccuracy.

My comment #15 was not amended, the manuscript is missing the part in which the authors show a clear connection between the results and the conclusions.

With regard to the authors answer to my comment #15, in Part 4.4 - what was the reference sample? Line 431 - "right most line" is not EPR terminology. Overall this explanation for signal intensity determination was not asked for in my comment #15,

We have corrected these inaccuracies, the answer is given in paragraph 10.

First, the relative intensity of the EPR spectrum of the complex Cu2+-(DETC)2 was calculated by the amplitude of its extreme high-field component of the hyperfine structure (HFS) (as the difference between the minimum and maximum of the first derivative of the absorption line).

In my opinion new Figure 7 is only suitable as a graphical abstract.

This figure is a graphic abstract. Whether to give it the status of a drawing is at the discretion of the editors and reviewers.

Title - Spectroscopic instead of Spectroscopy

In the new version of the title it is called so.

Comments on the Quality of English Language

Extensive editing of English language required. English must be improved prior to further peer review.

The English text was edited by a person who knows English well.

All these comments are taken into account in the new text of the article.

Sincerely,

Prof. Kh.L. Gainutdinov, Lieder scientist of Laboratory of Spin Physics and Spin Chemistry of Zavoisky Physical-Technical Institute of the Russian Academy of Sciences, professor of Kazan Federal University.